# Comparison of mental health and burnout between medical and nonmedical students

**Valerie Carrard**[1], **Céline Bourquin**[1], **Sylvie Berney**[2], **Pierre-Alexandre Bart**[3], **Patrick Bodenmann**[4], **Alexandre Berney**[1]*

1 Psychiatric Liaison Service, Department of Psychiatry, Lausanne University Hospital (CHUV) and University of Lausanne, Lausanne, Switzerland, 2 Service of General Psychiatry, Department of Psychiatry, Lausanne University Hospital (CHUV) and University of Lausanne, Lausanne, Switzerland, 3 Department of Internal Medicine, Lausanne University Hospital (CHUV) and University of Lausanne, Lausanne, Switzerland, 4 Chair of Medicine for Vulnerable Populations, University of Lausanne, Lausanne, Switzerland

* alexandre.berney@chuv.ch

## Abstract

The high rates of mental health issues and burnout among medical students have been well established. Some suggest that these high rates are due to medical studies being particularly demanding compared to other undergraduate training. However, research comparing the mental health and burnout of medical and nonmedical students yields inconsistent findings and is limited by small sample sizes as well as infrequent consideration of potential confounding risk factors. This study aimed to complement past research by comparing the mental health and burnout of medical students to those of the nonmedical students at the same university, while accounting for confounding risk factors. A total of 1057 medical and 870 nonmedical students participated in the study, completing validated questionnaires measuring mental health (depressive symptoms, suicidal ideation, anxiety) and burnout (emotional exhaustion, cynicism, academic efficacy), as well as 14 risk factors pertaining to sociodemographic (incl. deprivation), lifestyle, psychological characteristics, life stress, and social relations. After conducting t-tests to compare the two groups, the impact of the risk factors was assessed with adjusted regressions. Results revealed that medical students reported significantly *fewer* depressive symptoms, suicidal ideation, anxiety symptoms, and cynicism than nonmedical students. They also presented fewer of the examined risk factors. After adjusting for these factors, medical students still exhibited lower suicidal ideation and cynicism, but all other differences became non-significant. Moreover, when accounting for risk factors, a new significant difference appeared, with medical students presenting *more* emotional exhaustion than nonmedical students. These findings suggest that medical studies are not inherently more taxing than other undergraduate disciplines. Both medical and nonmedical students face significant mental health challenges, likely reflecting the typical strains of young adulthood, exacerbated by the pressures of demanding studies. The results

**Data availability statement:** The data that support the findings of this study are openly available in zenodo at http://doi.org/10.5281/zenodo.15149289.

**Funding:** This work is supported by the Swiss National Science Foundation (grant number 10001C_197442). The funders had no role in study design, data collection and analysis, decision to publish, or preparation of the manuscript.

**Competing interests:** The authors have declared that no competing interests exist.

underscore the need for holistic interventions to support the mental well-being of all undergraduate students, regardless of their discipline or faculty.

## Introduction

In the last few decades, there has been growing concern regarding the mental health and burnout of medical students. The fact that medical students present more mental health issues and burnout than the general population is well documented by reviews and meta-analyses [1–3]. A meta-analysis of meta-analyses indeed reports that 32.5% of medical students suffer from depression, 8.9% have suicidal thoughts, gestures, or acts, 32.5% present anxiety, and 35.8% exhibit burnout [4].

It has been suggested that these mental health issues and burnout might be due to the strains specific to medical studies. Indeed, medical studies are reported to be particularly challenging due to important workload, time pressure, stressful exams, competitive environment, relationship with hierarchy, co-workers, and patients during clinical rotations, as well as the confrontation with uncertainty, suffering, and death [5]. Studies focusing on nonmedical students are scarcer, but a systematic review has reported that nonmedical students also present a higher prevalence of depression compared to the general population [6]. To test whether the difficulties of medical studies would trigger more mental health issues and burnout than other types of curricula, several studies compared prevalences of mental health issues and burnout between medical and nonmedical students. However, the results are mixed. Some studies found that medical students present more depression [7,8] or burnout [7] than nonmedical students (physical education students or other college graduates), but a number of studies found no differences in depression [9–13], suicidal ideation [10,11], or anxiety [3,8] (compared to humanities, nursing, pharmacy, physical education, or business students), and others found that medical students present actually less depression [6, 14–17] or anxiety [9,17,18] than nonmedical students (humanities, nonmedical life-sciences, nursing, health sciences, architecture, English, law, or business students).

To shed light on the question, more studies in a single university setting, comparing more than two disciplines or faculties, and with a bigger sample size are needed. Also, it is essential to keep in mind that mental health and burnout are complex phenomena that can be influenced by numerous risk factors. Most past studies comparing the mental health or burnout of medical students to nonmedical students included gender and sometimes curriculum stage as control variables in order to isolate the specific impact of medical studies [3,6,13–18].

Sheldon and colleagues' [19] systematic review and meta-analysis of 66 longitudinal studies identified numerous other factors that significantly impact different indicators of students' mental health (depression, suicidal ideation, and other psychological distress indicators). Using Furber's validated taxonomy of mental illnesses' risk factors [20], they grouped them under 7 categories: *sociodemographic* (e.g., gender, socio-economic status), *physiological and health* (e.g., illness, sleep), *lifestyle* (e.g., physical activity), *psychological* (e.g., personality), *predictors of response to*

*trauma* (e.g., additional life stress), *relational* (e.g., social support), and *factors related to higher education* (e.g., academic environment). Importantly, these risk factors possibly differ between the medical and nonmedical student populations. To ascertain that it is the specificities of the studies that explain the potential differences in terms of mental health and burnout between medical and nonmedical students, research needs to test the concurrent impact of these non-academic factors that are known to put student at risk for mental health issues and burnout and that might differ between the medical and nonmedical students' population.

Therefore, the present study proposes to complement the existing literature by testing in a big sample whether medical students differ from all students in other faculties (i.e., social and political sciences, law, humanities, business, geoscience, pharmacy, biology, and theology students) within a single university. The first research question of the present study was thus: *do medical students differ from nonmedical students in terms of mental health (depressive symptoms, suicidal ideations, anxiety symptoms) and burnout (emotional exhaustion, cynicism, academic efficacy)?* Also, while it is virtually impossible to account for all the potential mental health risk factors identified in past literature, adjusting the analyses only for gender and curriculum stage as it has been done previously seems too limited to understand the impact of medical studies on such complex and multifaceted concept that is mental health or burnout. Thus, the second research question of the present study was: *are the differences in mental health and burnout between medical and nonmedical students primarily related to the nature of their studies, or do they stem from other underlying risk factors?* For the risk factors, the classification proposed by Sheldon was used. It includes sociodemographic, physiological, lifestyle, psychological, trauma-related, and relational risk factors.

## Methods

### Design and participants

This cross-sectional observational study is part of the ETMED-L project [21], a larger longitudinal project running from 2020 to 2024 that examines medical students' interpersonal competences and mental health with an open cohort design. The Research Ethics Committee of the Canton de Vaud approved the ETMED-L project, including the nonmedical students' data collection (project number 2020–02474) and all participants gave written informed consent.

### Medical students

As part of the ETMED-L project, an online questionnaire [22] was sent once a year to all medical students at the University of Lausanne (Switzerland), except external exchange students. The questionnaire took approximately one hour, and the students received 50CHF (≅50USD) for each completed questionnaire. The Lausanne Medical School has a six-year curriculum. Students from all six curriculum years were invited to participate, except in the last data collection, in which first-year students were excluded to favor longitudinal participation. Thus, the present study used the ETMED-L data of the 2022 questionnaire for the first-year medical students and that of the 2023 questionnaire for all other curriculum years (2–6). Each year the questionnaires were filled in from November 1 to December 2, which is an assignment-free and exam-free month for medical students.

Among the 1988 eligible medical students, 1138 participated and 1093 (53.17%) were included in the present study after exclusion of 45 students who gave a wrong answer to the attention questions placed in the questionnaire (e.g., 'In order to check your attention, please answer 'Slightly agree' to this question.').

### Nonmedical students

The ETMED-L project was exclusively designed for the study of medical students. However, towards the end of the project, the opportunity arose to survey nonmedical students at the university. Thus, a reduced version of the ETMED-L questionnaire was created. Compared to the original questionnaire for the medical student, the reduced version for the nonmedical students included the same instruments and scales to measure mental health, burnout and risk factors, but some

instruments measuring empathy, emotion recognition, and medical-studies-related information were taken out shortening its completion time form 1 hour to 15 minutes. It was then sent by email to all students from all faculties of the University of Lausanne, except the medical school (i.e., social and political sciences, law, humanities, business, geoscience, pharmacy, biology, and theology faculties). Note that the medical school is the only one with a six-year curriculum in Lausanne; all the others have a five-year curriculum, and all nonmedical students from all five curriculum years were invited to participate. Nonmedical students filled in their questionnaire from March 1 to April 2, 2024, an assignment-free and examination-free period of their curriculum. Since the ETMED-L project was designed for medical students and the nonmedical student survey was introduced later, no funds were allocated for the compensation of nonmedical students. Moreover, the questionnaire for nonmedical students was four times shorter, taking only 15 minutes to complete, whereas medical students'' questionnaires required approximately 1 hour. As a result, nonmedical students did not receive any compensation for their participation.

Among the 11536 eligible nonmedical students, 882 participated and 870 (7.54%) were included in the study after exclusion of 12 students who gave a wrong answer to the attention question.

## Measures

**Mental health and burnout.** The validated French versions of well-established instruments measuring three indicators of mental health and three dimensions of burnout were used in the present study. For mental health, *depressive symptoms* were measured with the Center for Epidemiological Studies-Depression (CES-D) [23,24]. A 20-item instrument assessing the occurrence of symptoms associated with depression over the past week. The CES-D showed a reliable prediction of depression among university students using cutoff scores of 16 for males, 20 for females, and 19 overall [25]. *Suicidal ideation* was measured with two items of the Beck Depression Inventory (BDI) assessing hopelessness and suicidal thoughts [26,27]. *Anxiety symptoms* were measured with the trait subscale of the State-Trait Anxiety Inventory (STAI), a 20-item instrument which measures the level of anxiety participants "generally feel" [28,29]. For burnout, the Maslach Burnout Inventory Student-Survey (MBI-SS) was used to assess the three dimensions of *emotional exhaustion* (5 items), *cynicism* (4 items), and *academic efficacy* (6 items, reversed dimension) [30,31].

**Mental health risk factors.** Sheldon and colleagues' [19] categorization was used to approach exhaustivity of risk factors considered, and included at least one indicator for each category that was not related to the academic environment. As a result, 14 risk factors were included in the present study. For the *Sociodemographic factors*, identifying as male (1 = male, 0 = female or non-binary), curriculum year, and the material deprivation index (i.e., financial and necessities deprivation) of the Deprivation in Primary Care Questionnaire (DiPCare-Q; [32]) were used. For the *Physiological and health factors*, the health deprivation index (i.e., presence of disabilities and chronic conditions impacting daily life) of the DiPCare [32] and the self-reported number of hours of sleep per night were measured. For *Lifestyle*, the number of hours of physical activity per week was assessed. For the *Psychological* factors, coping strategies (emotion-focused, problem-focused, and help-seeking strategies measured separately with the coping section of the Euronet questionnaire [33]) and self-reported satisfaction with health (from 1 = "very dissatisfied" to 5 = "very satisfied") were included. For the *Predictors of response to trauma*, which include negative life events, childhood trauma, and additional life stress [19], the number of hours spent in a paid job (on top of studies) per week was used as additional life stressor. For the *Relational* factors, the social deprivation index (i.e., isolation and lack of social activities) of the DipCare [32] and the availability of social support (emotional and practical support measured separately using an adaptation of the Swiss Household Panel items [34]) were assessed.

## Statistical analyses

After computing means and standard deviations for the overall, medical, and nonmedical samples, multiple imputation was applied (20 imputations) for the variables presenting an overall 5% of missing values or more using Multivariate Normal imputation. Variables with less than 5% missing data were not imputed, as studies indicate that multiple imputation yields minimal information gain at such low missing rates [35].

Then, independent sample t-tests were applied to examine whether and how the medical students differ from the nonmedical students regarding mental health (depressive symptoms, suicidal ideations, and anxiety symptoms), burnout (emotional exhaustion, cynicism, and academic efficacy), and the mental health risk factors included in the present study. *Cohen's d*s were computed to estimate effect size with *d*s of 0.2, 0.5, and 0.8 being considered respectively as small, medium, and large [36].

To test the impact of the mental health risk factors on the difference between medical and nonmedical students in terms of mental health and burnout, adjusted regressions with robust standard errors were modeled for each mental health indicator (depressive symptoms, suicidal ideations, and anxiety symptoms) and burnout dimension (emotional exhaustion, cynicism, and academic efficacy) separately with the samples (medical vs nonmedical students) as the independent variable and the 14 mental health risk factors as covariates. Standardized betas were reported with values of .10−.29 being considered as small, .30−.49 as medium, and .50 or greater as large [36]. According to the commonly recommended guidelines of having at least 10 observations per independent variable in a regression model, a minimum sample size of 150 is required to ensure adequate statistical power in the present study.

Additional sensitivity analyses were run. First, the t-tests and adjusted regressions were rerun with the complete non-imputed cases only to identify any potential impact of the multiple imputation applied. Then, given that the Lausanne Medical School is a six-year curriculum, whereas the other faculties have a five-year curriculum, the potential influence of this difference was tested by replicating the analyses while excluding the sixth-year medical students.

P-values of.05 were considered significant, and all analyses were run with Stata 18 [37].

## Results

### Sample descriptive

The sample descriptive are displayed in Table 1. The majority of the 1963 included students identified as being female (69.43%) with the proportion of students identifying as female being lower in the medical student sample (66.97%) than in the nonmedical one (72.53%). The students were on average 23 years old (range: 17–63) with the medical students being on average one year younger (22 years old, range: 17–52) than the nonmedical students (23 years old, range: 18–63). The distribution per curriculum year usually observed at the University of Lausanne was well reflected in our samples, with the first-year students representing around 25% of the samples and the rest of the students being almost equally spread across the other curriculum years.

Note that all the nonmedical disciplines of the University of Lausanne were represented in the 870 included nonmedical students with proportions that reflect well the sizes of each faculty: 316 from social and political sciences, 158 from law, 155 from humanities, 82 from business, 79 from geoscience, 71 from pharmacy or biology, and 9 from theology.

Of the 1963 included students, 1777 were complete cases and 186 contained missing data in at least one of the variables of interest (see Table 2). A Little's test indicated that the missing data could be considered as completely random in their occurrence (Chi$^2$ distance = 98.24, p = .137). After multiple imputation, missing values were reduced to 1.9%: only 5 medical students and 32 nonmedical students still had missing values on some of the non-imputed risk factors and were excluded from the analyses. The final analyzed sample thus included 1926 students, with 1088 medical students and 838 nonmedical students.

### Differences between medical and nonmedical students

The t-tests' results displayed in Table 2 showed that medical students reported significantly fewer mental health issues in all three indicators analyzed with small effect sizes for depressive symptoms and anxiety symptoms, and medium effect size for suicidal ideation. According to the CES-D cutoffs, 47.59% of the medical students are at risk of depression, whereas 55.19% of the nonmedical students can be considered at risk. Regarding burnout, the medical students reported

**Table 1. Sample descriptive.**

| | Overall | | Medical | | Nonmedical | |
|---|---|---|---|---|---|---|
| | Sample | | students | | students | |
| | (N = 1963) | | (N = 1093) | | (N = 870) | |
| | N | % | N | % | N | % |
| **Gender identification** | | | | | | |
| *Male* | 552 | 28.12 | 352 | 32.20 | 200 | 22.99 |
| *Female* | 1363 | 69.43 | 732 | 66.97 | 631 | 72.53 |
| *Non-binary* | 48 | 2.45 | 9 | 0.82 | 39 | 4.48 |
| **Curriculum year** | | | | | | |
| *Year1* | 512 | 26.08 | 274 | 25.07 | 238 | 27.36 |
| *Year2* | 324 | 16.51 | 148 | 13.54 | 176 | 20.23 |
| *Year3* | 309 | 15.74 | 164 | 15.00 | 145 | 16.67 |
| *Year4* | 328 | 16.71 | 170 | 15.55 | 158 | 18.16 |
| *Year5* | 354 | 18.03 | 201 | 18.39 | 153 | 17.59 |
| *Year6* | 136 | 6.93 | 136 | 12.44 | 0 | 0.00 |
| **Having a partner** | | | | | | |
| *Yes* | 949 | 48.34 | 420 | 48.28 | 529 | 48.40 |
| *No* | 1011 | 51.50 | 450 | 51.72 | 561 | 51.33 |
| *Unknown* | 3 | 0.15 | 0 | 0.00 | 3 | 0.27 |

also significantly less cynicism with a small effect size but did not differ from the nonmedical students in terms of emotional exhaustion nor academic efficacy.

Overall, medical students also presented significantly fewer mental health risk factors than nonmedical students. First, they differed significantly in their gender identification, with the proportion of students identifying as male being higher in the medical students' sample (32.20%) than in the nonmedical students' sample (22.99%). Also, the medical students were younger, presented less material deprivation, reported less health deprivation, slept fewer hours per night, had higher satisfaction with their health, used more adaptive coping strategies (less emotion-focused, more problem-focused, and more help-seeking coping), and reported having less social deprivation as well as more emotional and practical social support available than nonmedical students. All these differences were significant with p values < .01, but Cohen's $d$s indicated small effect sizes at best.

## Impact of mental health risk factors

The regressions' results displayed in Table 3 show that adjusting for risk factors modifies substantially the differences between medical and nonmedical students in terms of mental health and burnout. Regarding mental health, differences in depressive and anxiety symptoms were largely accounted for by the included risk factors. Specifically, the effect size for depressive symptoms became negligible ($\beta = 0.04$), and the difference in anxiety symptoms was no longer significant after adjustment. Suicidal ideation was the only mental health indicator that remained significantly associated with the type of studies, with medical students reporting less suicidal ideation. However, the effect size decreased from medium to small after controlling for the risk factors. For burnout, cynicism remained significant with a small effect size, suggesting that medical studies are associated with lower levels of cynicism compared to nonmedical studies, independent of other mental health risk factors. Furthermore, differences in emotional exhaustion that were otherwise not significant became significant when mental health risk factors are adjusted for, with medical students presenting more emotional exhaustion with a small effect size. The same is also observed for academic efficacy (reversed dimension of burnout) with medical

**Table 2. T-tests comparing medical students to nonmedical students.**

| | Overall sample (N = 1963) | | | Medical students (N = 1093) | | Nonmedical students (N = 870) | | t-tests (N after MI = 1926) | | |
|---|---|---|---|---|---|---|---|---|---|---|
| | %Missing | Mean | SD | Mean | SD | Mean | SD | t | p | Cohen's d |
| **Mental Health** | | | | | | | | | | |
| *Depressive symptoms* | 8.15 | 20.16 | 11.63 | 18.99 | 11.35 | 21.90 | 11.84 | −5.35 | <.001 | 0.25 |
| *Suicidal ideation* | 8.41 | 0.88 | 1.20 | 0.65 | 1.03 | 1.23 | 1.35 | −10.37 | <.001 | 0.50 |
| *Anxiety symptoms* | 9.02 | 47.56 | 11.97 | 45.98 | 11.92 | 49.97 | 11.65 | −7.24 | <.001 | 0.34 |
| **Burnout** | | | | | | | | | | |
| *Emotional exhaustion* | 9.32 | 16.56 | 5.19 | 16.50 | 4.91 | 16.66 | 5.59 | −0.79 | .460 | 0.03 |
| *Cynicism* | 9.32 | 10.22 | 4.65 | 9.48 | 4.29 | 11.36 | 4.95 | −8.67 | <.001 | 0.41 |
| *Academic efficacy* | 9.32 | 23.94 | 4.67 | 24.00 | 4.53 | 23.85 | 4.90 | 0.78 | .471 | 0.03 |
| **Risk factors** | | | | | | | | | | |
| *Identifying as male* | 0.00 | 0.28 | 0.45 | 0.32 | 0.47 | 0.23 | 0.42 | 4.53 | <.001 | 0.20 |
| *Curriculum year* | 0.00 | 3.05 | 1.65 | 3.26 | 1.76 | 2.78 | 1.46 | 6.42 | <.001 | 0.29 |
| *Material deprivation* | 1.32 | 1.13 | 1.55 | 1.01 | 1.43 | 1.30 | 1.68 | −4.09 | <.001 | 0.19 |
| *Health deprivation* | 1.88 | 0.30 | 0.56 | 0.23 | 0.51 | 0.40 | 0.62 | −6.76 | <.001 | 0.30 |
| *Sleep hours per day* | 0.25 | 7.12 | 0.94 | 7.03 | 0.92 | 7.22 | 0.97 | −4.41 | <.001 | 0.20 |
| *Physical activities* | 0.25 | 3.42 | 2.94 | 3.51 | 2.86 | 3.31 | 3.02 | 1.49 | .138 | 0.07 |
| *Satisfaction with health* | 0.25 | 3.61 | 0.99 | 3.73 | 0.97 | 3.45 | 0.99 | 6.21 | <.001 | 0.29 |
| *Emotion-focused coping* | 9.37 | 10.12 | 4.14 | 9.57 | 3.99 | 10.96 | 4.23 | −6.98 | <.001 | 0.34 |
| *Problem-focused coping* | 9.37 | 7.17 | 1.77 | 7.27 | 1.69 | 7.02 | 1.87 | 2.99 | .003 | 0.14 |
| *Help-seeking coping* | 9.48 | 5.33 | 2.83 | 5.57 | 2.88 | 4.96 | 2.71 | 4.46 | <.001 | 0.22 |
| *Hours in paid job per week* | 0.15 | 4.22 | 7.53 | 2.96 | 6.62 | 5.79 | 8.28 | −8.44 | <.001 | 0.38 |
| *Social deprivation* | 1.88 | 0.50 | 0.76 | 0.45 | 0.70 | 0.57 | 0.81 | −3.54 | <.001 | 0.16 |
| *Emotional social support* | 9.48 | 8.48 | 2.04 | 8.62 | 1.98 | 8.27 | 2.12 | 3.51 | <.001 | 0.17 |
| *Practical social support* | 9.48 | 7.67 | 2.32 | 7.83 | 2.31 | 7.43 | 2.32 | 3.54 | <.001 | 0.17 |

*Note.* MI = Multiple imputation. Cohen's *d*s of 0.2, 0.5, and 0.8 are considered, respectively, as small, medium, and large [36].

students presenting significantly less academic efficacy than nonmedical students, but the effect size indicates a negligible difference.

## Sensitivity analyses

The t-tests and adjusted regressions replicated with the complete cases showed no differences in any of the results (see S1 and S2 Tables). Results of the analyses excluding the sixth-year medical students showed also no difference to the one including all curriculum years (see S3 and S4 Tables).

## Discussion

Comparing a large sample of medical students to nonmedical students from all other faculties in a single university, the present study confirms that medical students present elevated rates of mental health issues, but further indicates that these high rates are also present among nonmedical students. Indeed, going against the general belief that medical studies are associated with more distress than other disciplines or faculties, the present study showed that medical students actually reported significantly fewer depressive symptoms, suicidal ideations, anxiety symptoms, and cynicism

Table 3. Adjusted regressions testing the difference between medical and nonmedical students (N = 1926).

| | Mental health | | | | | | | | | Burnout | | | | | | | | |
|---|---|---|---|---|---|---|---|---|---|---|---|---|---|---|---|---|---|---|
| | Depressive symptoms | | | Suicidal ideation | | | Anxiety symptoms | | | Emotional exhaustion | | | Cynicism | | | Academic efficacy | | |
| | β | SE | p | β | SE | p | β | SE | p | β | SE | p | β | SE | p | β | SE | p |
| Medical students | .04 | 0.42 | .041 | −.12 | 0.05 | <.001 | −.01 | 0.41 | .550 | .10 | 0.22 | <.001 | −.13 | 0.22 | <.001 | −.07 | 0.22 | .001 |
| Identifying as male | −.06 | 0.45 | .001 | .03 | 0.06 | .170 | −.07 | 0.45 | <.001 | −.05 | 0.26 | .029 | .06 | 0.25 | .022 | −.03 | 0.25 | .183 |
| Curriculum year | −.09 | 0.12 | <.001 | −.04 | 0.01 | .040 | −.07 | 0.12 | <.001 | −.07 | 0.07 | .002 | .14 | 0.07 | <.001 | −.02 | 0.06 | .404 |
| Material deprivation | .07 | 0.14 | <.001 | .02 | 0.02 | .398 | .03 | 0.13 | .072 | .06 | 0.08 | .008 | .05 | 0.08 | .084 | .00 | 0.07 | .843 |
| Health deprivation | .19 | 0.46 | <.001 | .20 | 0.06 | <.001 | .14 | 0.41 | <.001 | .14 | 0.20 | <.001 | .06 | 0.22 | .019 | −.06 | 0.21 | .023 |
| Sleep hours per day | −.10 | 0.21 | <.001 | −.05 | 0.03 | .022 | −.10 | 0.21 | <.001 | −.11 | 0.12 | <.001 | −.01 | 0.11 | .763 | .05 | 0.11 | .015 |
| Physical activities | .00 | 0.07 | .986 | .00 | 0.01 | .938 | −.01 | 0.07 | .494 | −.07 | 0.04 | .001 | .01 | 0.04 | .787 | .03 | 0.04 | .165 |
| Satisfaction with health | −.18 | 0.22 | <.001 | −.09 | 0.03 | <.001 | −.16 | 0.23 | <.001 | −.14 | 0.12 | <.001 | −.12 | 0.12 | <.001 | .11 | 0.12 | <.001 |
| Emotion-focused coping | .39 | 0.06 | <.001 | .24 | 0.01 | <.001 | .49 | 0.05 | <.001 | .31 | 0.03 | <.001 | .25 | 0.03 | <.001 | −.25 | 0.03 | <.001 |
| Problem-focused coping | .01 | 0.12 | .701 | −.03 | 0.02 | .244 | −.01 | 0.11 | .399 | −.01 | 0.06 | .744 | −.06 | 0.06 | .006 | .18 | 0.06 | <.001 |
| Help-seeking coping | −.08 | 0.08 | <.001 | −.08 | 0.01 | <.001 | −.05 | 0.08 | .014 | −.03 | 0.04 | .163 | −.07 | 0.04 | .005 | .08 | 0.04 | .001 |
| Hours in paid job per week | .00 | 0.03 | .982 | .01 | 0.00 | .571 | .00 | 0.03 | .780 | .00 | 0.01 | .983 | .03 | 0.01 | .138 | −.01 | 0.01 | .571 |
| Social deprivation | .06 | 0.28 | .002 | .07 | 0.04 | .001 | .02 | 0.28 | .178 | .06 | 0.16 | .013 | −.02 | 0.15 | .331 | .03 | 0.16 | .335 |
| Emotional social support | −.06 | 0.13 | .004 | −.06 | 0.02 | .046 | −.04 | 0.12 | .070 | .02 | 0.07 | .581 | −.06 | 0.07 | .055 | .08 | 0.07 | .009 |
| Practical social support | −.11 | 0.11 | <.001 | −.11 | 0.02 | <.001 | −.11 | 0.11 | <.001 | −.05 | 0.06 | .078 | −.02 | 0.06 | .541 | .09 | 0.06 | .002 |
| F | 144.91 | | | 51.27 | | | 184.85 | | | 57.40 | | | 29.32 | | | 33.78 | | |
| F's p-value | <.001 | | | <.001 | | | <.001 | | | <.001 | | | <.001 | | | <.001 | | |
| R2 | .55 | | | .33 | | | .57 | | | .32 | | | .20 | | | .22 | | |

Notes. Betas of .10−.29 were considered small, .30−.49 medium, and .50 or greater large [36].

than nonmedical students. Similar results were reported in some past studies showing that medical students present less depression [6,14–17] and anxiety [9,17,18] than nonmedical students.

However, the results of the present study show that the differences are generally small and that most of them are due to mental health risk factors that do not relate to the studies per se. Indeed, the medical students' sample presented a higher proportion of individuals identifying as male, less material, health, and social deprivation, more adaptive coping strategies, and better social support. When controlling for these risk factors in order to isolate the specific impact of the studies per se, the differences observed in terms of depressive and anxiety symptoms became negligible or non-significant. Also, a difference in terms of emotional exhaustion appeared, suggesting that medical studies trigger more emotional exhaustion than nonmedical studies, but this impact can only be seen when adjusting for confounding factors, because it is counterbalanced by lesser risk factors and more protective factors among medical students. In summary, even though the results of the present study indicate some significant differences between medical and nonmedical students in terms of mental health and burnout, it does not indicate that medical studies are more or less taxing than other studies. As indicated by the adjusted regressions' results, medical students are less at risk for mental health issues and burnout than the other students primarily because they represent a different type of population that is characterized by fewer biopsychosocial disadvantages.

As already mentioned, the past literature is mixed regarding the differences between medical and nonmedical students in terms of mental health and burnout. Some found that medical students present more mental health issues and burnout [7,8], whereas others found the opposite [9,6,14–18] or no significant difference [3,8–13]. In a sense, the results of the present study align with all this past literature as they indicate that one can find a significant difference in one direction or

the other, or even no significant differences, depending on which indicator of mental health and burnout is examined and which risk factors are adjusted for or not. Therefore, future research on mental health among university students needs to include several indicators of mental health and account for potential confounding risk and protective factors to develop a comprehensive understanding of the issues involved.

Many reasons might explain why medical students present systematically fewer biopsychosocial disadvantages than nonmedical students. This includes differences in financial family support that could facilitate studies reputed as more demanding, better educational opportunities among less deprived populations, sociocultural emphasis favoring medicine in less deprived contexts, higher self-esteem among less deprived individuals, less developed social networks among more deprived populations, and admission regulations that could favor less deprived applicants (e.g., requiring citizenship or permanent residency). According to the regressions' results, these different factors seem to have an impact on the overall observed mental health. There is thus a clear need for more services to support students facing biopsychosocial disadvantages, and especially for nonmedical students. Most universities, including the University of Lausanne, already propose these kinds of services such as social services for students in precarious situations, psychotherapeutic and spiritual support to care for the students' psychological health, varied students' association and social events to increase the social support available to students, as well as on campus sport facilities and healthy food options to favor healthy living. Regarding the material deprivation issues, university tuitions are already very affordable in Switzerland (i.e., the equivalent of around 1100 USD/year at the University of Lausanne). Reasonable tuition could be further supplemented by more affordable living facilities and scholarships.

Perhaps the most significant implication of this study's findings is that both medical and nonmedical students face considerable mental strain typical of young adults, exacerbated by the additional challenges of demanding undergraduate studies. Indeed, between 48% and 55% of the present study's students can be considered at risk of depression according to the CES-D validated cutoff. According to a sensitivity/specificity analysis reported elsewhere [38], this would correspond to around 17% of the students presenting an actual clinical depression. These numbers call for particular attention to the mental health issues of all university students across all disciplines and faculties through easily accessible and affordable psychotherapeutic services.

## Limitations

While the findings of this study provide valuable insights into the mental health and burnout of medical and nonmedical students, several limitations should be considered when interpreting the results. First, the medical students' data were collected in November, whereas the nonmedical students' data were collected in March. While both are assignment-free and exam-free months, they are still at different stages of the academic year and in different seasons. However, the medical curricula structure is drastically different from those of other faculties, no matter the year-period. Also, the findings actually indicate the reverse of the usually observed seasonal effect, with the students surveyed in winter showing fewer mental health issues and burnout than those surveyed in spring. Thus, surveying the two populations in the same month would likely have increased the differences reported in the present study.

Second, the sample of nonmedical students demonstrated a significantly lower participation rate and a higher amount of missing data compared to the medical student sample. This discrepancy can be attributed to the ETMED-L project being exclusively designed to study medical students and the nonmedical student survey being a later addition to the project. Thus, all the communication and advertising around the project was primarily focused on medical students. Also, the medical students were financially compensated for their participation, while nonmedical students were not, because the survey for nonmedical students was significantly shorter than the one completed by medical students and no funds were allocated for this part of the project. As a result, more participation biases can be expected within the nonmedical students' sample. Indeed, individuals presenting more socioeconomical or health-related vulnerabilities (including mental health issues) are known to participate less in cohort studies [39] and might be more underrepresented in the nonmedical

sample than in the medical sample. However, a higher participation rate of more vulnerable individuals in our nonmedical sample would actually increase the differences observed in the present study. Also, it would potentially counterbalance the under-representation of male participants in our nonmedical sample, a population known to be less interested in participating in health research [40] and reporting generally fewer mental health issues. Nonetheless, the size of the nonmedical sample remains large, and the covariates included in the regression analyses enabled to account for several potential participation biases (i.e., gender-, socio-economical-, and health-related ones).

Third, interpretations of the results related to suicidal ideation should be approached with caution, as it was not the focus of the present paper and was thus assessed with only two items. These items were not specifically validated as a comprehensive measure of suicidal ideation and may not capture its full complexity.

Finally, by accounting for the differences between medical and nonmedical students in terms of sociodemographic, physiological, lifestyle, psychological, life stress, and relational factors, the present study strived to isolate the specific impact of medical studies. However, it is virtually impossible to isolate entirely the impact of the studies themselves, because it is impossible to consider every other possible influencing factor. Nevertheless, the present study goes a step further compared to past research that controlled only for gender and sometimes curriculum stage. Giving that it is impossible to isolate the impact of specific studies and also impossible to manipulate it, future research should favor qualitative approaches to explore the specificities of the different studies in order to reach a comprehensive understanding of the impact of specific studies on mental health and burnout.

## Conclusion

The main takeaway of the present study is that mental health and burnout rates are rather high for all university students, no matter their disciplines or faculties, and that it warrants interventions. The study's results indeed indicate that medical studies cannot be considered more or less taxing than other studies. Services and interventions aimed at improving the mental wellbeing of university students should thus be developed in a holistic way that considers both the complexities of undergraduate training as well as multiple external and internal resources and aggravating factors.

## Supporting information

**S1 Table. T-tests comparing medical students to nonmedical students based on complete cases.**
(PDF)

**S2 Table. Adjusted regressions testing the difference between medical and nonmedical students based on complete cases (N = 1,777).**
(PDF)

**S3 Table. T-tests comparing medical students to nonmedical students while excluding sixth-year medical students.**
(PDF)

**S4 Table. Adjusted regressions testing the difference between medical and nonmedical students while excluding sixth-year medical students (N = 1790).**
(PDF)

## Author contributions

**Conceptualization:** Céline Bourquin, Alexandre Berney.

**Data curation:** Valerie Carrard.

**Formal analysis:** Valerie Carrard.

**Funding acquisition:** Alexandre Berney.

**Methodology:** Valerie Carrard.

**Project administration:** Céline Bourquin, Alexandre Berney.

**Supervision:** Céline Bourquin, Alexandre Berney.

**Writing – original draft:** Valerie Carrard.

**Writing – review & editing:** Valerie Carrard, Céline Bourquin, Sylvie Berney, Pierre-Alexandre Bart, Patrick Bodenmann, Alexandre Berney.

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
