## [Decision Letter · Decision Letter 0]

30 Jul 2025

Dear Dr. Berney,

Thank you for submitting your manuscript to PLOS ONE. After careful consideration, we feel that it has merit but does not fully meet PLOS ONE’s publication criteria as it currently stands. Therefore, we invite you to submit a revised version of the manuscript that addresses the points raised during the review process.

We look forward to receiving your revised manuscript.

Kind regards,

Yaser Mohammed Al-Worafi

Academic Editor

PLOS ONE

Journal Requirements:

Reviewers' comments:

Reviewer's Responses to Questions

**Comments to the Author**

1. Is the manuscript technically sound, and do the data support the conclusions?

Reviewer #1: No

Reviewer #2: Partly

2. Has the statistical analysis been performed appropriately and rigorously?

Reviewer #1: No

Reviewer #2: Yes

3. Have the authors made all data underlying the findings in their manuscript fully available?

Reviewer #1: No

Reviewer #2: Yes

4. Is the manuscript presented in an intelligible fashion and written in standard English?

Reviewer #1: Yes

Reviewer #2: No

Reviewer #1: This is an interesting study about the prevalence of mental symptoms and burnout in a huge sample of medical students (n=1.057) (in comparison with non medical students, n=870). The article is well written, used a correct methodology and realize multivariate analysis in order to adjust the results for several risk factors. For us the principal problem of the article is in the methods section. Authors should explain if they are studying a possible bias selection.

1) We have some questions about how the sample was enrolled. Why are there more participants in the medical students sample? Why the participation rate is so different (53% versus 7%) in each group?

2) Why the questionnaire for medical students is different to non medical? What are these differences? How can compare the researches the results between the two groups if they are not using the same scales? Please include more details about that important point.

3) Why the missing data are more frequent in non medical sample in comparison con medical sample?

4) Why the medical students receive a monetary incentive and the non medical do not receive anything? Can this circumstance bias the results?

5) In some variables (as for example psychological factors) the researches analyze important questions (as coping strategies) that are difficult to study using a questionnaire alone

6) As the authors confirm in the discussion section the principal result of the study (the medical students have better mental health and less burnout than nonmedical) is against the principal results exposed in the introduction section. I wonder if the possible selection bias can explain, at least in part, these differences.

7) Some of the arguments used in the discussion section (for example when they enumerate the reasons that can explain why medical students present systematically less biopsychosocial disadvantages) are speculative and are not supported by the fundings if this study.

8) Unfortunately the study is not very interesting for the reader because do not answer the second aim of this study: what are the specific risk factors (or protective factors) associated with mental health in medical students.

Reviewer #2: While the manuscript is interesting, I believe the survey design is a major limitation. I appreciate that you cannot address this as the study is complete but it would need further emphasis in the limitation section as the very least. See details below

The language and grammar should be checked thoroughly throughout the document.

Some of the paragraphs are very long. It would be good to break them up.

Abstract

Reword first line. ‘Important’ rates doesn’t read well. Perhaps use The Elevated rates….

‘Seldom consideration’, also needs to be reworded

Introduction

Consider rephrasing the first sentence.

In the first paragraph comparisons are made between medical students and the general population. It would be good to include some references about comparison between students in general and the general population to set the scene.

Methodology

It is not good practice to use pronouns. For example, state that Sheldon’s et al. was used… rather than ‘we’. Check throughout and remove pronouns.

Details with regards to the sample would fit better under in the methodology section, when detailing the participants in the study.

Details about the attention questions should also be included under the methodology.

Results

A demographics table would be useful, outlining gender, age, curriculum year etc.

Why did you include different risk factors rather than keeping it consistent?

-Age was used as Risk Factor in the T-Test but Curriculum Year was used in the Regression

Consider re-running the analyses for consistency.

Limitations

The design of the study is a major limitation to the study and should be considered as such.

The different timings of survey collection has been noted as a limitation. There may be different pressures at these different time points, particularly later in the academic year when assignments are due and examinations are pending (non-medical students). Non-compensation of the non-medical students may also have an impact, as those with mental health problems may have been more likely to take part in the survey, as they may be more invested, which may partially account for the high rates of problems in this cohort. Furthermore, the nonmedical students were much more likely to be female, who consistently report higher levels of mental health problems such as depression and anxiety in research studies. These factors also need to be noted as limitations.

**Do you want your identity to be public for this peer review?** For information about this choice, including consent withdrawal, please see our Privacy Policy

Reviewer #1: No

Reviewer #2: No

---

## [Author Response · Author response to Decision Letter 1]

10 Sep 2025

Comparison of mental health and burnout between medical and nonmedical students

PONE-D-25-33531

- Point-by-point response to the Editorial office and the reviewers’ comments -

Dear Professor Al-Worafi,

Thank you for the opportunity to resubmit our work for publication consideration in PLOS ONE. We carefully addressed all points raised by both the Editorial office and the reviewers. We hope that our point-by-point responses and the revisions made to the manuscript will resolve all the concerns raised and that our paper now meets the standards for publication.

Journal Requirements:

- The manuscript has been carefully checked, and the tables’ format was modified to meet PLOS ONE's style requirements.

- The ethics statement was deleted from the Structured disclosure section and now only appears in the Methods section.

- The funding information was deleted from the manuscript.

- NA (no recommendation to cite specific work)

Reviewer #1

Comment:

This is an interesting study about the prevalence of mental symptoms and burnout in a huge sample of medical students (n=1.057) (in comparison with non medical students, n=870). The article is well written, used a correct methodology and realize multivariate analysis in order to adjust the results for several risk factors. For us the principal problem of the article is in the methods section. Authors should explain if they are studying a possible bias selection.

Response:

We thank reviewer #1 for their insightful comments on our Methods section. We would like to start with a preliminary statement that could clarify many aspects of the study design. The ETMED-L project was actually exclusively designed for medical students. It is only towards the end of the project, that we had the opportunity to survey nonmedical students at the university. We thus adapted the medical student questionnaire by reducing the number of measures included (while keeping the one retained identical) and sent it by email to all nonmedical students at our university. Consequently, communication and advertising for the ETMED-L project always targeted medical students. Also, no funds were allocated for the nonmedical students’ survey and the questionnaire for the nonmedical students was significantly shorter with a completion time of 15 minutes compared to a 1-hour questionnaire for the medical students. Thus, the nonmedical students received no compensation for their participation, whereas medical student received 50CHF when fully completing the questionnaire.

Some of these details regarding our sample and design were mentioned only briefly in the previous version of our manuscript, and we now recognize the need for greater precision. As described in our responses to your different comments, we have expanded the Methods and Limitation sections and hope that these revisions have satisfactorily improved the technical soundness of our work.

Comment:

1) We have some questions about how the sample was enrolled. Why are there more participants in the medical students sample? Why the participation rate is so different (53% versus 7%) in each group?

Response:

Thank you for this comment. We hope that our preliminary statement now clarifies how our samples were enrolled. As indicated, the Methods section has been significantly revised and now better describe the fact that the nonmedical survey was a later addition to a project targeting medical students and the differences in terms of compensation:

“The ETMED-L project was exclusively designed for the study of medical students. However, towards the end of the project, the opportunity arose to survey nonmedical students at the university.” (lines 107-109)

“Since the ETMED-L project was designed for medical students and the nonmedical student survey was introduced later, no funds were allocated for the compensation of nonmedical students. Moreover, the questionnaire for nonmedical students was four times shorter, taking only 15 minutes to complete, whereas medical students' questionnaires required approximately 1 hour. As a result, nonmedical students did not receive any compensation for their participation.” (lines 120-125)

These aspects of the study design can indeed explain the difference in terms of response rates, because the nonmedical students were not targeted by the project’s advertisement and were not compensated for their participation. Also, the medical survey was longitudinal, likely fostering a long-term relationship with participants, enhancing their sense of loyalty and willingness to participate. The limitation section now describes how these aspects could have impacted the response rates:

“Second, the sample of nonmedical students demonstrated a significantly lower participation rate and a higher amount of missing data compared to the medical student sample. This discrepancy can be attributed to the ETMED-L project being exclusively designed to study medical students and the nonmedical student survey being a later addition to the project. Thus, all the communication and advertisement around the project was primarily focused on medical students. Also, the medical students were financially compensated for their participation, while nonmedical students were not, because the survey for nonmedical students was significantly shorter than the one completed by medical students and no fund was allocated for this part of the project.” (lines 304-312)

Comment:

2) Why the questionnaire for medical students is different to non medical? What are these differences? How can compare the researches the results between the two groups if they are not using the same scales? Please include more details about that important point.

Response:

Thank you for giving us the opportunity to clarify this point. The scales included in the nonmedical students’ questionnaire were identical to those in the medical students' questionnaire, allowing for direct comparison between the two groups. Indeed, the medical students' questionnaire was just edited to retain only the measures deemed essential, creating a shorter version for the nonmedical students. For instance, the medical student’s questionnaire included measures of emotion recognition, professional identity, and stress that were not included in the nonmedical students’ questionnaire. The exact correspondence of the scales used for medical and nonmedical students is now indicated in our Methods section as follows:

“a reduced version of the ETMED-L questionnaire was created. Compared to the original questionnaire for the medical student, the reduced version for the nonmedical students included the same instruments and scales to measure mental health, burnout and risk factors, but some instruments measuring empathy, emotion recognition, and medical-studies-related information were taken out shortening its completion time form 1 hour to 15 minutes.” (lines 109-113)

Comment:

3) Why the missing data are more frequent in non medical sample in comparison con medical sample?

Response:

The higher rate of missing data in the nonmedical students is due to a greater number of students not completing their questionnaire to the end. This might primarily be due to the nonmedical students not receiving any compensation for the completion of the questionnaire, whereas the medical students received a compensation if they filled in the questionnaire entirely. We hope that the revision made to the limitation section (lines 305-314), as outlined in our response to your comment 1, clarifies this point.

Comment:

4) Why the medical students receive a monetary incentive and the non medical do not receive anything? Can this circumstance bias the results?

Response:

Thank you for raising this issue. We hope that our preliminary statement and the revisions made to our Methods section clarify the reasons for the difference in participant compensation. The difference in compensation between medical and nonmedical students may indeed have introduced selection biases that impact response rates and participant representation. Compensation for medical students was designed to encourage participation from populations that typically engage less in cohort and health-related studies, such as males [1] and individuals facing socio-economic deprivation, physical health issues, or mental health challenges [2]. If nonmedical students had also received compensation, we might have seen a higher representation of individuals with physical and mental health issues, which would actually amplify the differences in mental health outcomes reported in our results. On the other hand, this effect might be counterbalanced by a larger proportion of male participants, who generally report fewer mental health concerns. Consequently, while selection biases may exist, their overall impact on the results is likely minimal, as the effects could offset each other. This issue has been elaborated upon in the Limitation section of our study as follows:

“As a result, more participation biases can be expected within the nonmedical students’ sample. Indeed, individuals presenting more socioeconomical or health-related vulnerabilities (including mental health issues) are known to participate less in cohort studies [41] and might be more underrepresented in the nonmedical sample than in the medical sample. However, a higher participation rate of more vulnerable individuals in our nonmedical sample would actually increase the differences observed in the present study. Also, it would potentially counterbalance the under-representation of male participants in our nonmedical sample, a population known to be less interested in participating in health research [40] and reporting generally fewer mental health issues. Nonetheless, the size of the nonmedical sample remains large, and the covariates included in the regression analyses enabled to account for several potential participation biases (i.e. gender-, socio-economical-, and health-related ones).” (lines 312-322)

1. Borg DJ, Haritopoulou-Sinanidou M, Gabrovska P, Tseng HW, Honeyman D, Schweitzer D, et al. Barriers and facilitators for recruiting and retaining male participants into longitudinal health research: A systematic review. BMC Med. Res. Methodol. 2024;24:46.

2. Rothenbühler M, Voorpostel M. Attrition in the Swiss Household Panel: Are Vulnerable Groups more Affected than Others? In: Oris M, Roberts C, Joye D, Ernst Stähli M, editors. Surveying Human Vulnerabilities across the Life Course. Cham: Springer International Publishing; 2016. page 223–44.

Comment:

5) In some variables (as for example psychological factors) the researches analyze important questions (as coping strategies) that are difficult to study using a questionnaire alone

Response:

Indeed, we acknowledge that complex psychological factors are challenging to fully capture through self-reported questionnaires. Therefore, a qualitative approach would be a valuable avenue to complement the findings of this study. Nonetheless, in a quantitative setting, coping strategies questionnaires similar to the one used in the present study have been shown to be consistently related to meaningful outcomes such as resilience and psychological wellbeing among university students (see for instance: [1]).

1. Elisabetta Sagone, and Maria Elvira De Caroli, “A Correlational Study on Dispositional Resilience, Psychological Well-being, and Coping Strategies in University Students.” American Journal of Educational Research, vol. 2, no. 7 (2014): 463-471. doi: 10.12691/education-2-7-5

Comment:

6) As the authors confirm in the discussion section the principal result of the study (the medical students have better mental health and less burnout than nonmedical) is against the principal results exposed in the introduction section. I wonder if the possible selection bias can explain, at least in part, these differences.

Response:

Past literature exposes mixed results regarding the differences between medical and nonmedical students in terms of mental health and burnout with some showing that medical students have poorer mental health and others finding the reverse pattern or no significant differences. As indicated in our response to your comment no 4, it is unlikely that the reported results are solely attributable to selection biases. Indeed, if nonmedical students had also received compensation, we might have seen a higher representation of individuals with physical and mental health issues, which would likely amplify the observed differences in mental health outcomes. Yet, this potential increase could be counterbalanced by a larger proportion of male participants, who typically report fewer mental health concerns. Thus, while selection biases may exist, their overall impact on the results is likely minimal.

Comment:

7) Some of the arguments used in the discussion section (for example when they enumerate the reasons that can explain why medical students present systematically less biopsychosocial disadvantages) are speculative and are not supported by the fundings if this study.

Response:

In the Discussion section, we strive to propose possible explanations as interpretations of the observed findings. Naturally, these interpretations extend beyond the results themselves, as they seek to open avenues for reflections, practical implications, and future research. However, they are always grounded in the actual findings of the study and existing literature. We thoroughly reviewed the entire section to ensure that each interpretation is derived from the actual findings of the study or existing literature.

Comment:

8) Unfortunately the study is not very interesting for the reader because do not answer the second aim of this study: what are the specific risk factors (or protective factors) associated with mental health in medical students.

Response:

We believe our work holds significant value as it addresses a clear gap in the literature, which has presented mixed findings on the topic, likely due to the infrequent consideration of potential influencing factors. We have rephrased our second research question for clarity, because our aim was not to identify the factors associated with medical students’ mental health per se, but to investigate the factors that could explain the potential differences between medical and nonmedical students in terms of mental health. We hope that this new formulation will provide Reviewer #1 with a clearer perspective on the significance of the study.

“Thus, the second research question of the present study was: are the differences in mental health and burnout between medical and nonmedical students primarily related to the nature of their studies, or do they stem from other underlying risk factors?” (lines 78-81)

Reviewer #2:

Comment:

While the manuscript is interesting, I believe the survey design is a major limitation. I appreciate that you cannot address this as the study is complete but it would need further emphasis in the limitation section as the very least. See details below

Response:

We thank you for your insightful feedback on our manuscript that helped us enhance its overall clarity. In response to Reviewer

---

## [Editor Report · Decision Letter 1]

12 Sep 2025

Comparison of mental health and burnout between medical and nonmedical students

PONE-D-25-33531R1

Dear Dr. Berney,

We’re pleased to inform you that your manuscript has been judged scientifically suitable for publication and will be formally accepted for publication once it meets all outstanding technical requirements.

Kind regards,

Yaser Mohammed Al-Worafi

Academic Editor

PLOS ONE

---

## [Editor Report · Acceptance letter]

PONE-D-25-33531R1

PLOS ONE

Dear Dr. Berney,

I'm pleased to inform you that your manuscript has been deemed suitable for publication in PLOS ONE. Congratulations! Your manuscript is now being handed over to our production team.

Kind regards,

on behalf of

Professor Yaser Mohammed Al-Worafi

Academic Editor

PLOS ONE